# Multiscale real time and high sensitivity ion detection with complementary organic electrochemical transistors amplifier

Paolo Romele [1], Paschalis Gkoupidenis [2], Dimitrios A. Koutsouras[2], Katharina Lieberth[2], Zsolt M. Kovács-Vajna [1], Paul W. M. Blom [2] & Fabrizio Torricelli [1✉]

Ions are ubiquitous biological regulators playing a key role for vital processes in animals and plants. The combined detection of ion concentration and real-time monitoring of small variations with respect to the resting conditions is a multiscale functionality providing important information on health states. This multiscale functionality is still an open challenge for current ion sensing approaches. Here we show multiscale real-time and high-sensitivity ion detection with complementary organic electrochemical transistors amplifiers. The ion-sensing amplifier integrates in the same device both selective ion-to-electron transduction and local signal amplification demonstrating a sensitivity larger than 2300 mV V$^{-1}$ dec$^{-1}$, which overcomes the fundamental limit. It provides both ion detection over a range of five orders of magnitude and real-time monitoring of variations two orders of magnitude lower than the detected concentration, viz. multiscale ion detection. The approach is generally applicable to several transistor technologies and opens opportunities for multifunctional enhanced bioelectronics.

[1] University of Brescia, Department of Information Engineering,  via Branze 38, 25123 Brescia, Italy. [2] Max Planck Institute for Polymer Research, Ackermannweg 10, 55128 Mainz, Germany. ✉email: fabrizio.torricelli@unibs.it

ons are the ubiquitous biological and physiological regulators, being involved in most of the fundamental processes of every living organism. Ions enable the communication between cells and the vital metabolic and bioenergetic processes, playing a key role in hydration, pH regulation, and osmotic balance across cell membranes[1,2]. Ions regulate the operation of nerves, muscles, and neural signals in animals and the activation of enzymes, turgor, and photosynthesis in plants. As a consequence, life is permeated by a close connection with ions, and the ion concentration levels in biological systems can give important information on their state of health. For instance, an optimal ionic uptake of plants from the soil is fundamental to exploit their vital functions, but ionic excesses can be toxic[3,4]. Analogously, abnormal levels in human body fluids are often the fingerprint of ongoing pathological states and diseases[5], organ disfunctions such as heart or kidney failure, and dehydration[6]. Ion regulation in living organisms involves intra and extracellular fluctuations from the resting condition, which typically ranges from $10^{-4}$ M to $10^{-1}$ M, and small deviations from the optimal equilibrium levels can be associated with pathological states[7]. As an example, the normal potassium concentration in human serum is in the range from 3.5 $10^{-3}$ to 5.5 $10^{-3}$ M and a departure from this range lower than 20% can be associated with severe morbidity and mortality[8]. Analogously, during an epileptic seizure the neuronal extracellular concentration of potassium ($K^+$) increases from 4 $10^{-3}$ M to 12 $10^{-3}$ M while the sodium ($Na^+$) concentration decreases from 145 $10^{-3}$ M to 139 $10^{-3}$ M[9]. Therefore, a multiscale approach able to detect the ion concentrations equilibria as well as their deviation from equilibrium conditions is very relevant in biological applications. This would be an invaluable tool for the study and diagnosis of pathological states in biological systems and, more in general, for several emerging fields including environmental monitoring, water control and test, healthcare, agriculture, food, and drink industries.

So far, ion detection and monitoring have been widely addressed with several transistor-based technologies, comprising silicon, zinc-oxide, and graphene ion-sensitive field-effect transistors[10-13], porous silicon extended-gate FETs[14], amorphous indium-gallium-zinc-oxide dual-gate thin-film transistors[15], organic electrolyte-gated FETs[16] and organic electrochemical transistors (OECTs)[17-19]. Among the aforementioned approaches, OECTs are gaining significant interest because they combine superior performance in terms of sensitivity and low-voltage operation, with the typical features of organic technologies, viz. stable operation in aqueous environment, mechanical softness, biological compatibility, ease of integration into arrays, low-cost and large-area fabrication[20-22]. OECTs rely on ionic-electronic charge interaction, which is obtained by putting the electrolyte in direct contact with the electronic channel and the subnanometric ionic-electronic interaction involves the whole microscale volume of the transistor channel[23]. This volumetric response results in an outstanding ion-to-electron transduction.

OECT-based ion sensors[17-19,24] mostly focus on a material or transistor architecture approach, and the resulting sensitivity is defined by the fundamental Nernst limit equal to 59 mV dec$^{-1}$[25]. This is very far from the ideal sensitivity limit, which can be set to about 1000 mV dec$^{-1}$ assuming the maximum supply voltage for stable operation in aqueous environment, viz. 1 V, and a minimum ion concentration range of one decade. To improve the sensitivity, we recently proposed a current-driven OECT architecture, achieving selective $K^+$ sensitivity up to 414 $10^{-3}$ V dec$^{-1}$ over an ion concentration range $10^{-2}$ M $-$ 1 M at a supply voltage equal to 0.8 V[26]. However, although sensitive this approach requires a read-out scheme not ideally suitable for real-time ion sensing[27]. Therefore, while OECTs are a promising technology for ion detection and real-time monitoring, current approaches

still require additional electronic circuits for signal amplification because of the limited sensitivity. Even more importantly, the possibility to detect both the ion concentration in a wide range and track small variations of the ion concentration with respect to the detected concentration – viz. multiscale high-sensitivity ion detection – is still an open challenge.

Here we show simultaneous multiscale real-time and high-sensitivity ion detection with complementary OECTs integrated in a push-pull amplifier configuration. The ion sensing amplifier provides both ion-to-electron transduction and signal amplification demonstrating selective real-time ion detection with a sensitivity up to 1172 mV dec$^{-1}$ at a supply voltage equal to 0.5 V and with an operative range of $10^{-5}$ M–1 M. This results in a voltage sensitivity normalized to the supply voltage larger than 2300 mV V$^{-1}$ dec$^{-1}$, the highest value ever reported. The high sensitivity allows to monitor variations of the concentration of physiologically relevant ions in human blood serum lower than 20%, making the complementary OECT amplifier a suitable technology for the detection of pathological states through ion sensing. Importantly, we demonstrate that the performances can be enhanced and easily tuned by means of simple design parameters. The high sensitivity combined with the multifunctional reconfiguration enable both ion detection over a range of five orders of magnitude and real-time monitoring of variations two orders of magnitude lower than the detected concentration, viz. multiscale high-sensitivity ion detection.

## Results

**Ion sensitive OECT complementary amplifier.** The ion sensitive OECT amplifier consists of a p-type and an n-type OECT connected in series (Fig. 1a). This circuit configuration is a universal building block used in digital electronic circuits to implement the NOT logic function[28-32]. Here we use this circuit topology following an unexplored OECT-aware analog design approach, where local transduction and amplification are integrated in the same device. Moreover, the use of the complementary configuration results in enhanced gain and reduced power consumption compared with unipolar resistive-based OECT amplifiers used for electrophysiological signal amplification[33,34].

The prototypical conductive polymer poly (3,4-ethylenedioxythiophene) doped with poly(styrene sulfonate) (PEDOT:PSS) is used for the p-type OECT and poly (benzimidazobenzophenanthroline) (BBL) is used for the n-type OECT[32] (details on the fabrication are provided in the "Methods" section). Both polymers show very good stability in aqueous environment, are commercially available, and the corresponding OECTs display comparable ON currents (Supplementary Fig. 1). The polymers show top performing electrical properties among the OECT materials[35], enabling the integration of high performance OECT amplifiers.

A typical transfer characteristic $V_O$-$V_I$ of the fabricated ion sensitive OECT complementary amplifier is shown in Fig. 1b. The characteristic displays a rail-to-rail output voltage swing with five distinct regions, which correspond to different operating conditions of the p-type and n-type OECTs. More in detail, when $V_I = 0$ V, the n-type OECT is turned off, whereas the p-type OECT is highly conductive and works in the linear regime, resulting in $V_O = V_{DD}$ (Fig. 1b, region 1, $V_I < 0.2$ V). By increasing $V_I$, the pull-up action of the p-type OECT becomes weaker while the pull-down of the n-type OECT increases and, as a consequence, the output voltage decreases (region 2, 0.2 V $< V_I$ $< 0.3$ V). Interestingly, when $V_I \approx 0.3$ V the output voltage $V_O$ shows a sharp and large variation (region 3). This transition region is very narrow and extends of only few millivolts around the transition voltage $V_M$, which we define as the input voltage $V_I$

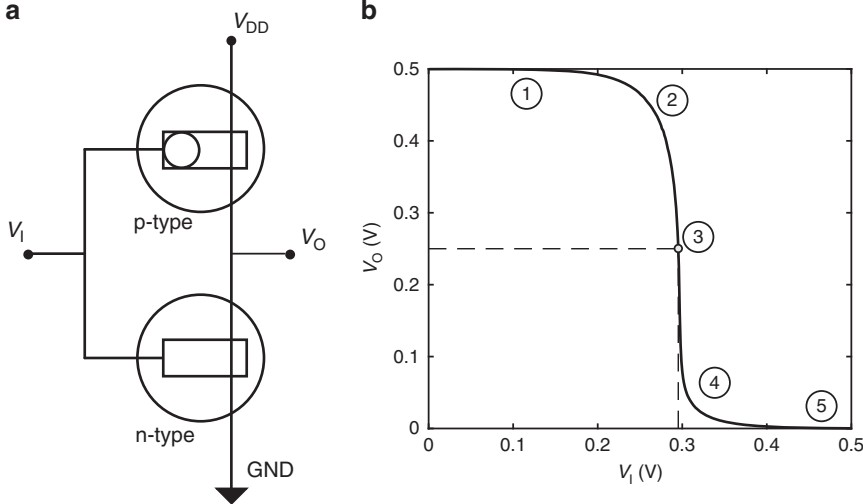

**Fig. 1 Organic electrochemical transistor complementary amplifier. a** Organic electrochemical transistor complementary amplifier architecture. The input voltage ($V_I$) is connected to the gates of the transistors, and the output voltage ($V_O$) is collected at their drains. The supply voltage ($V_{DD}$) is connected to the source of the p-type OECT, and the source of the n-type is grounded (GND, $V_{GND} = 0$ V). **b** Typical transfer characteristic of an OECT complementary amplifier. The gray dot identifies $V_M$ (viz. the required $V_I$ to have $V_O = V_{DD}$ /2) and the various regions of operation are highlighted with numbers.

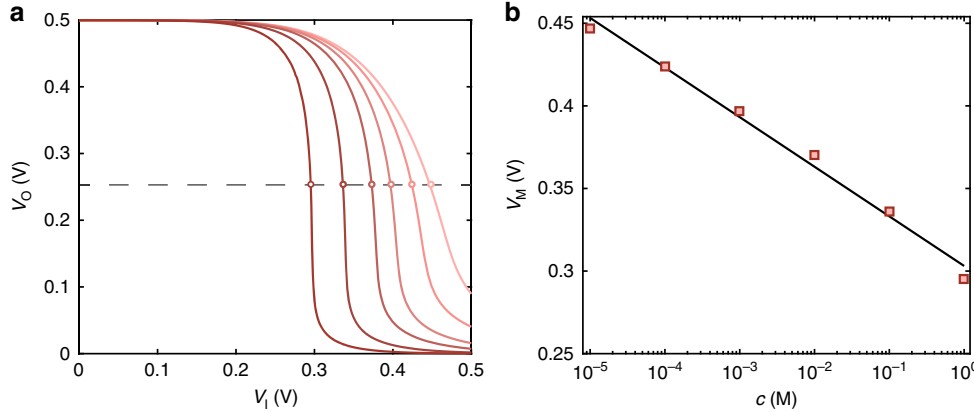

**Fig. 2 Wide range operation. a** Measured (thick red lines) transfer characteristics of the organic electrochemical transistor complementary amplifier at various ion concentrations. From the rightmost to the leftmost characteristic the ion concentration is equal to [$10^{-5}$, $10^{-4}$, $10^{-3}$, $10^{-2}$, $10^{-1}$, 1] M. **b** Measured $V_M$ as a function of $c$ (square symbols). A sensitivity equal to $S_M = 30$ mV dec$^{-1}$ is obtained with the least square linear approximation (solid line) of the measurements.

required to obtain $V_O = V_{DD}/2$ (gray dot). In other words, close to $V_M$, small variations of $V_I$ result in large variations of $V_O$, thus providing a large amplification $A = \mathrm{d}V_O/\mathrm{d}V_I$. It is worth to note that this amplification is inherently related to the circuit configuration that enables the simultaneous high-gain operation (saturation regime) of both p-type and n-type OECTs. In this regard, OECTs are the ideal technology for this amplifier circuit architecture because they provide both large transconductance $g_m$[36] and output resistance $r_o$[37], which eventually result in a large intrinsic gain equal to $g_m r_o$[38]. When $V_I$ is few millivolts larger than $V_M$, the p-type OECT still operates in the saturation regime while the n-type OECT switches to the linear regime (region 4, $0.3$ V $< V_I < 0.4$ V). By further increasing $V_I$ the output voltage approaches GND (region 5, $V_I > 0.4$ V).

**Wide range ion detection**. To investigate the ion response of the OECT-based complementary amplifier, we measured the $V_O$-$V_I$ characteristics as a function of the ion concertation $c$ in the range $10^{-5}$ M to 1 M. Figure 2a shows that the measured electrical characteristics (full lines) systematically shift and the transition

voltage $V_M$ (marked with dots in Fig. 2a) reduces by increasing the ion concentration. Figure 2b shows $V_M$ as a function of $c$. $V_M$ decreases by increasing $c$ with a sensitivity $S_M = \partial V_M/\partial c$ equal to 30 mV dec$^{-1}$ over five orders of magnitude of ion concentration. This can be explained by considering the ion sensitive properties of the OECTs combined with the circuit architecture. More in detail, Lin et al.[25] showed that the measured transfer curves of PEDOT:PSS OECTs shift to more negative gate voltages by increasing the ion concentration. Recently, we demonstrated that this is due to the fixed charges in the polyelectrolyte phase of the polymer, which results in an ion concentration dependent voltage drop at the electrolyte/polyelectrolyte interface and, as a consequence, a threshold voltage shift of the OECT electrical characteristics is measured[39].

The OECT threshold voltage $V_T$ can be related to $V_M$ by considering the OECT complementary amplifier configuration and using the Bernards–Malliaras OECT drain current model[40], $V_M$ reads (see Supplementary Eq. (5), Supplementary Note 1):

$$V_M = \frac{V_{Tn} + \eta V_{Tp} + \eta V_{DD}}{1 + \eta}, \tag{1}$$

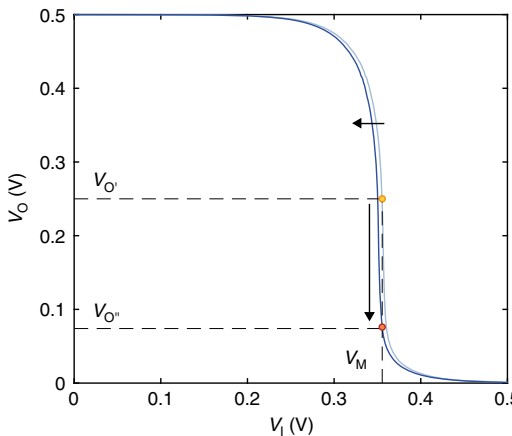

**Fig. 3 Ionic-to-electronic amplification mechanism.** To operate the organic electrochemical transistor complementary amplifier with high sensitivity, the input voltage is biased at $V_I = V_M$, while the output voltage is continuously measured. Light-blue characteristic: at $V_I = V_M$ the output $V_O = V_{O'}$ and the ion concentration is $c_0$. A small variation of the ion concentration results in a shift of the electrical characteristic (from light-blue to dark-blue) and this, in turn, in a large variation of the output voltage from $V_{O'}$ to $V_{O''}$.

where $V_{Tn}$, $V_{Tp}$ are the threshold voltages of the p- and n-type OECTs[39], respectively, $\eta = (\Gamma_p \Gamma_n^{-1})^{1/2}$, and $\Gamma_p$, $\Gamma_n$ are the current pre-factors of the p- and n-type OECTs, respectively. Equation (1) shows that $V_M$ depends on the OECTs threshold voltages $V_{Tn}$, $V_{Tp}$, and on the designed OECT geometries (viz. $\Gamma_p$ and $\Gamma_n$). Basing on Eq. (1) the sensitivity $S_M$ can be calculated:

$$S_M = \frac{\partial V_M}{\partial c} = \frac{\eta}{1 + \eta} \frac{\partial V_{Tp}}{\partial c}, \qquad (2)$$

where in the case of PEDOT:PSS OETCs $\partial V_{Tp}/\partial c = 59$ mV dec$^{-1}$[39]. We designed the OECTs in order to have comparable currents, resulting in $\eta \approx 1$ and hence $S_M = 0.5 \ \partial V_{Tp}/\partial c$. We note that the ion sensitivity of $V_M$ is smaller than the OECT threshold voltage sensitivity. As a consequence, $V_M$ obtained from the transfer characteristics $V_O$-$V_I$ enables ion detection in a wide range ($10^{-5}$ M – 1 M) with a supply voltage as low as $V_{DD} = 0.5$ V.

**Local ion-to-voltage amplification.** Upon ion detection in a wide range, the OECT complementary amplifier is used to detect small variations of the ion concentration with high sensitivity. The previously recorded $V_M$ yields the corresponding ion concentration $c_0$ according to the calibration curve showed in Fig. 2b. The input voltage $V_I$ is biased at $V_I = V_M$ and in this operating condition, owing to the amplification $A$ of the OECT complementary amplifier configuration, a very small variation of ion concentration results in a large variation of the output voltage $V_O$. As an example, Fig. 3 shows that $V_O$ varies from $V_{O'} = 0.25$ V (orange dot) to $V_{O''} = 0.08$ V (red dot) in the case the ion concentration variation $\Delta c = c - c_0 = 50 \ 10^{-3}$ M and $c_0 = 10^{-1}$ M. More in detail, the output voltage decreases when $c$ increases with respect to the initial concentration $c_0$, while a larger $V_O$ is displayed when the ion concentration gets lower. The output voltage variation $\partial V_O$ resulting from a concentration variation $\partial c$ can be quantified as follows:

$$S_A = \frac{\partial V_O}{\partial c} = A S_M, \qquad (3)$$

where $S_M$ is given by Eq. (2) and $A$ can be calculated by differentiating the OECT current equation (see Supplementary Eq.

(10), Supplementary note 1) with respect to $V_I$ and results:

$$A(V_M) = \frac{1}{I_D(V_M)} \frac{g_{mn}\left[1 + \lambda_n \frac{V_{DD}}{2}\right] + g_{mp}\left[1 + \lambda_p \frac{V_{DD}}{2}\right]}{\lambda_n + \lambda_p}, \qquad (4)$$

where $I_D(V_M)$ is the current flowing through the OECTs at $V_I = V_M$ and $\lambda_n$, $\lambda_p$ account for the channel length modulation[41] in the n-type and p-type OECTs, respectively. Interestingly, Eq. (4) shows that the OECT complementary amplifier leverages on the combined large transconductance (viz. large $g_{mn}$, $g_{mp}$) and output resistance (viz. small $\lambda \propto g_o$, $g_o = r_o^{-1}$) of OECTs to provide, at the same time, enhanced ion-to-voltage transduction and amplification, namely high-sensitivity ion detection. Indeed, OECTs show extremely high $g_m$[36] and $r_o$[37] owing to the volumetric ionic-electronic charge interaction, making them ideal components for the fabrication of high-performance ion sensitive amplifiers.

**Real-time high-sensitivity operation.** The real-time, high-sensitivity monitoring of the ion concentration is shown in Fig. 4a. The input voltage of the OECT complementary amplifier is biased at $V_I = V_M$ and the output voltage $V_O$ is measured over the time while the analyte ion concentration is varied from $54 \ 10^{-5}$ M to $1.27 \ 10^{-3}$ M. It is worth to note that a variation of $c$ as small as $7 \ 10^{-5}$ M, from $54 \ 10^{-5}$ to $61 \ 10^{-5}$, results in an output voltage response $\Delta V_O = 20$ mV with a response time of only 11 s (Supplementary Fig. 2). Interestingly, such variation is detected with respect to a background concentration of about one order of magnitude larger ($c_0 = 54 \ 10^{-5}$). Figure 4b shows the measured steady-state output voltage as a function of $c$. The least square linear approximation of the measured $V_O$ provides an average sensitivity equal to 640 mV dec$^{-1}$ (Fig. 4b).

The high sensitivity characteristics of the OECT complementary amplifier for various ranges of ion concentrations are displayed in Fig. 4c. We obtained an average voltage sensitivity equal to 640 mV dec$^{-1}$, 837 mV dec$^{-1}$, 913 mV dec$^{-1}$, and 1124 mV dec$^{-1}$, when $c$ is in the range $5 \ 10^{-4}$–$10^{-3}$ M, $5 \ 10^{-3}$–$10^{-2}$ M, $5 \ 10^{-2}$–$10^{-1}$ M and $5 \ 10^{-1}$–$10^0$ M, respectively. To easily compare the various ion-to-voltage responses of the OECT complementary amplifier, the variation of the output voltage $\Delta V_O$ as a function of $c$ normalized to the minimum concentration of the assessed range (viz. $c/c_{min}$) is displayed in Fig. 4d. At a supply voltage $V_{DD} = 0.5$ V the maximum $\Delta V_O$ is about 0.2 V in the lowest range of concentration and it increases to almost 0.3 V in the highest range of concentration. This confirms that the voltage sensitivity increases by increasing the ion concentration range, and this can be explained by considering that the amplification parameter $A$ increases with increasing $c$ (see Supplementary Note 2 and Supplementary Fig. 3). This is an important feature to detect small variations of $c$ even when operating at large equilibrium or background concentrations. More in detail, the minimum ion variation that can be detected is $\Delta c_{min} = [10^{\wedge}(\Delta V_{O,min} S_A^{-1}) - 1] c_{max}$ (see Supplementary Note 2), where $c_{max}$ is the maximum concentration in the operative range, $S_A$ is the sensitivity and $\Delta V_{Omin}$ is the corresponding minimum output voltage variation. To be conservative, we assume $\Delta V_{Omin} = 1\% \ V_{DD}$, viz. in our case $\Delta V_{Omin} = 5 \ 10^{-3}$ V, and this gives $\Delta c_{min} = 1.8 \ 10^{-5}$ M, $1.4 \ 10^{-4}$ M, $1.3 \ 10^{-3}$ M, $1.0 \ 10^{-2}$ M when $c_{max} = 10^{-3}$ M, $10^{-2}$ M, $10^{-1}$ M, $10^0$ M, respectively. Therefore, the OECT complementary amplifier provides $\Delta c_{min} / c_{max} \cong 1\%$ over the whole assessed range (viz. $5 \ 10^{-4}$–$10^0$ M). This performance perfectly fits the most challenging physiological applications that require the detection of ion variations of the order of $10^{-3}$ M when the ion concentration is in the range $10 \ 10^{-3}$–$100 \ 10^{-3}$ M[7–9].

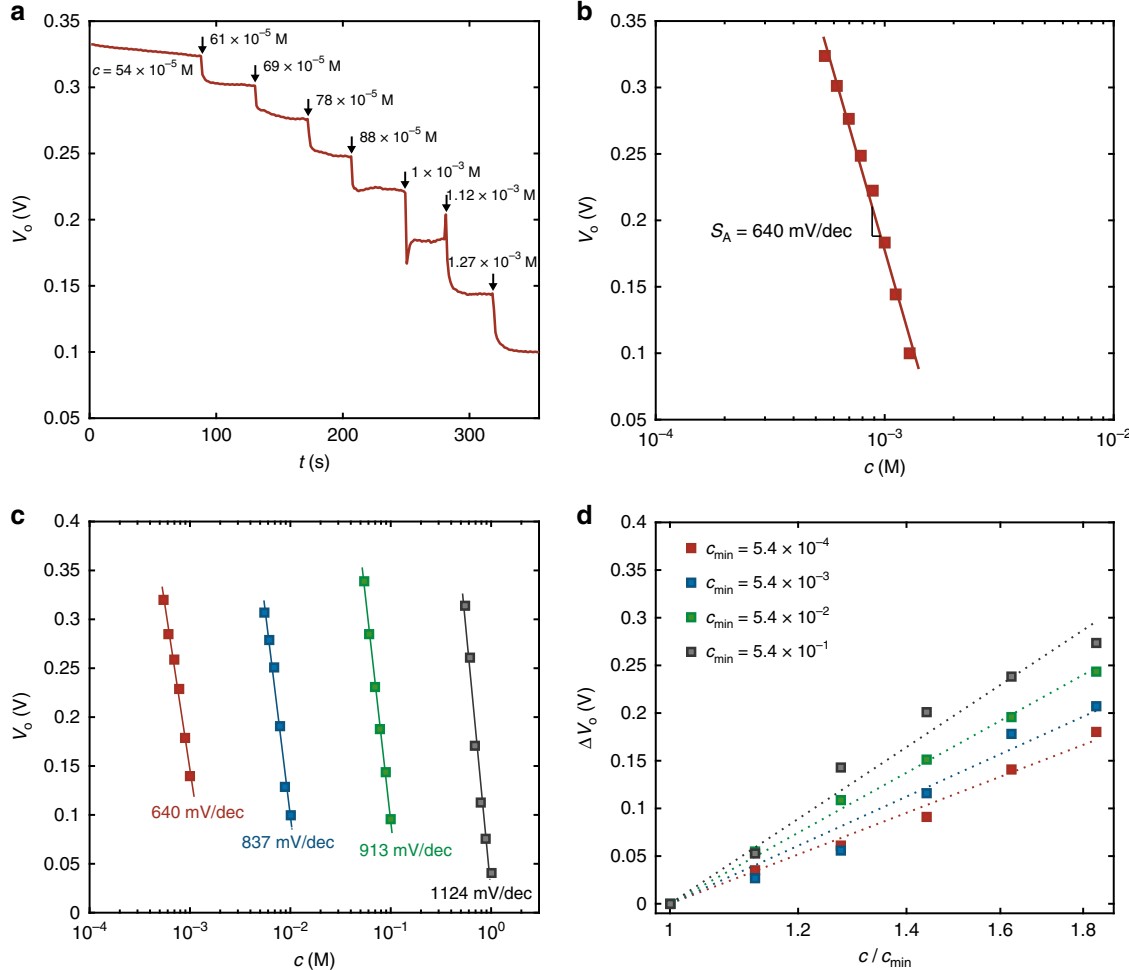

**Fig. 4 Real time high sensitivity operation. a** Real time high-sensitivity ion detection in the concentration range $54 \times 10^{-5}$ M – $1.27 \times 10^{-3}$ M. The electrolyte concentration is increased every 30 s. The measurements are performed at $V_M = 0.415$ V. **b** Measured steady-state output voltage $V_O$ (symbols) as a function of ion concentration $c$. Full red line is the linear least square fit to the measurements and yields a sensitivity $S_A = 640$ mV dec$^{-1}$. **c** Measured $V_O$ as a function of $c$ in various sub-range of concentrations, covering the whole physiological range. The ion concentration where $V_O = V_{DD} / 2$ is $c_0 = [7.8 \times 10^{-4}, 7.8 \times 10^{-3}, 7.8 \times 10^{-2}, 7.8 \times 10^{-1}]$ M. Full lines are the linear least square fit to the measurements, yielding $S_A$. **d** Measured (symbols) output voltage variation $\Delta V_O = V_O(c) - V_O(c_{min})$ as a function of $c / c_{min}$, where $c_{min} = [5.4 \times 10^{-4}, 5.4 \times 10^{-3}, 5.4 \times 10^{-2}, 5.4 \times 10^{-1}]$ M. Dotted lines are guides for the eye.

**Tuning of the ion sensitive performances.** In contrast to single transistor approaches where the performances are imposed by the device physics, the proposed circuit-oriented approach allows to tune the ion sensing performances by design of the circuit parameters. Importantly, the understanding of the OECT amplifier provided in the previous sections gives precise information on the design parameters in order to tailor the target figures of merit and performance. As a relevant example, here we show that the sensitivity of the OECT complementary amplifier can be easily tuned to meet specific requirements of various application fields, which is important to broad its use. According to Eq. (4), the ion sensitivity depends on both technological and circuit parameters ($g_{mn}$, $g_{mp}$, $\lambda_n$, $\lambda_p$) and on the supply voltage $V_{DD}$. Therefore, the design of the OECTs parameters allows to set the ion detection performances at the fabrication stage, while the dependence of the sensitivity on $V_{DD}$ enables dynamic electrically-reconfigurable ion sensing response. Figure 5 displays the output voltage variation as a function of $c/c_{min}$ when the OECT amplifier is operated at several $V_{DD}$. The required sensitivity and subrange of ion concentration can be finely tuned by simply changing $V_{DD}$. More in detail, when $c_{min} = 5 \ 10^{-2}$, a supply voltage of 0.5 V allows to detect concentration variations

with a sensitivity of 913 mV dec$^{-1}$ up to $c = 1.8 \ c_{min}$ (dark green squares). By lowering the $V_{DD}$ to 0.4 V the sensitivity reduces to 483 mV dec$^{-1}$, extending the sub-range to $c = 2.8 \ c_{min}$ (green circles), while at $V_{DD} = 0.3$ V the sensitivity is equal to 267 mV dec$^{-1}$ up to $c = 3.8 \ c_{min}$ (light green triangles).

**Multiscale ion-selective operation.** Ion selectivity using the OECT complementary amplifier is obtained by integrating ion selective membranes (ISMs). Analogously to the ion-selective sensors based on organic transistors[16,17,19], ISM is placed between the gate and the channel. The ISMs are fabricated according to the methods reported in Ref. [17] (see "Methods" section). As a relevant application, we show K$^+$-selective OECT complementary amplifiers. K$^+$ plays a fundamental role in human body, being involved in the regulation of intracellular water uptake, in the transmission of neural signals and in muscles contraction[1]. The electrical characteristics of the K$^+$-selective OECT complementary amplifier are measured as a function of the K$^+$ concentration $c^{K+}$. Figure 6a displays $V_M$ as a function $c^{K+}$ in the range from $10^{-5}$ to 1 M. It shows that $V_M$ decreases with increasing $c^{K+}$ in agreement with the measurements without the K$^+$-selective membrane (Fig. 2b).

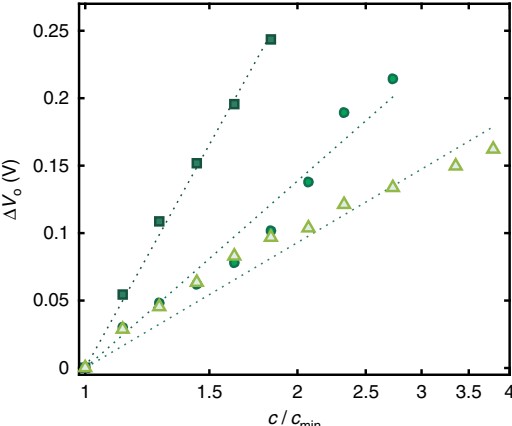

**Fig. 5 Tuning of the amplifier sensitivity.** Measured (symbols) output voltage variation $\Delta V_O = V_O(c) - V_O(c_{min})$ by varying the operating voltage $V_{DD}$. The dark green squares are measured at $V_{DD} = 0.5\,V$ and $c$ equal to [5.4, 6.1, 6.9, 7.8, 8.8, 10] x $10^{-2}$ M, the green circles are measured at $V_{DD} = 0.4\,V$ and $c$ equal to [5.4, 6.1, 6.9, 7.8, 8.8, 10, 11.3, 12.7, 14.9, 18.3] x $10^{-2}$ M and the light green triangles are measured at $V_{DD} = 0.3\,V$ and $c$ equal to [5.4, 6.1, 6.9, 7.8, 8.8, 10, 11.3, 12.7, 14.9, 18.3, 20.6] x $10^{-2}$ M. Dotted lines are guides for the eye.

Upon selective $K^+$ detection over wide range, the input voltage of the OECT complementary amplifier is biased at $V_I = V_M(K^+)$ and $V_O$ is measured as a function of time by varying the $K^+$ concertation into the analyte solution. Figure 6b shows that the measured $V_O$ decreases with increasing $K^+$ concentration and the linear least square approximation of the measured $V_O$ as a function of $K^+$ provides an average sensitivity equal to 703 mV $dec^{-1}$, 952 mV $dec^{-1}$, 995 mV $dec^{-1}$, and 1172 mV $dec^{-1}$, when $K^+$ is in the range 5 $10^{-4}$–$10^{-3}$ M, 5 $10^{-3}$–$10^{-2}$ M, 5 $10^{-2}$–$10^{-1}$ M, and 5 $10^{-1}$–$10^0$ M, respectively. It is worth to note that the measured sensitivity is in full agreement with the sensitivity obtained without the ion selective membrane (Fig. 4c) demonstrating that the use of ISM does not reduce the OECT amplifier sensitivity.

The selectivity of the ion selective OECT amplifier against other physiologically relevant ions is displayed in Fig. 6c–f. In Fig. 6c the $K^+$ concentration is constant and amounts to 5.4 $10^{-2}$ M, while the $Na^+$ concentration is systematically increased from 5.4 $10^{-2}$ M to 1 $10^{-1}$ M. The measured output voltage $V_O$ as function of time is independent of the $Na^+$ spiked to the electrolyte solution, as confirmed by Fig. 6d where the measured steady state $V_O$ is displayed as a function of the $Na^+$ concentration. A negligible response is triggered by the $Na^+$ concentration and the cross-sensitivity amounts to 33 mV $dec^{-1}$, which is about 30-folds lower than the $K^+$ sensitivity in the same range of concentrations (i.e., 995 mV $dec^{-1}$) and 20-folds lower than the minimum $K^+$ sensitivity (i.e., 703 mV $dec^{-1}$ in the range 5 $10^{-4}$ – 1 $10^{-3}$ M) (Supplementary Fig. 4). As a further confirmation, we assessed the selectivity against $Ca^{2+}$, a relevant bivalent ion present in physiological systems. The $Ca^{2+}$ concentration is systematically increased from 5.4 $10^{-2}$ M to 1 $10^{-1}$ M and, as showed in Fig. 6e, the measured $V_O$ as function of time is independent of the $Ca^{2+}$ spiked to the electrolyte solution. Figure 6f shows the measured steady state $V_O$ as a function of the $Ca^{2+}$ concentration. A negligible response is triggered by the $Ca^{2+}$ concentration and the cross-sensitivity amounts to 20 mV $dec^{-1}$, which is about 50-folds lower than the $K^+$ sensitivity in the same range of concentrations (i.e., 995 mV $dec^{-1}$) and 35-fold lower than the minimum $K^+$ sensitivity (i.e., 703 mV $dec^{-1}$ in the range 5 $10^{-4}$–1 $10^{-3}$ M) (Supplementary Fig. 4). Therefore, we can conclude that the

ion selective OECT complementary amplifier architecture dramatically enhances the sensitivity of ISM-based OECT ion sensors while ensuring selective response.

**Blood serum ion monitoring.** As practical and feasible application of the complementary OECT amplifier in the biomedical field, high-sensitivity and selective ion detection in blood serum is demonstrated. Figure 7a shows the real-time monitoring of potassium in blood serum with the complementary OECT amplifier. The input voltage is biased at 0.37 V and the output voltage is continuously measured over time. The analyte is human blood serum, with a starting potassium concentration of 4.8 $10^{-3}$ M and a background sodium concentration of 1.36 $10^{-1}$ M, i.e., about 2 orders of magnitude larger than the $K^+$ concentration. The output voltage is continuously measured over time by varying the potassium concentration from 4.8 $10^{-3}$ M to 9.6 $10^{-3}$ M, which are $K^+$ concentrations representative of heathy and pathological states. We note that the $K^+$ concentration of the blood serum is varied by spiking the analyte with a small amount of 1 $10^{-1}$ M KCl solution. Figure 7a shows that when the analyte $K^+$ concentration is increased, the output voltage readily and systematically decreases. The measured steady state $V_O$ as a function of $c$ is shown in Fig. 7b. The least square linear approximation of the $V_O$-$c$ characteristic yields an average sensitivity equal to 662 mV $dec^{-1}$, which is in agreement with that obtained in the case of KCl water solution in the same range of $c$. The selectivity control experiment is displayed in Fig. 7c, where the starting sodium concentration of 1.36 $10^{-1}$ M is increased over time reaching up to 2.72 $10^{-1}$ M by spiking with a 5 M NaCl solution. A negligible variation of the measured output voltage is displayed. We note that the maximum $Na^+$ concentration assessed in this experiment is larger than the maximum value reached in humans' blood[42]. Figure 7d shows the measured steady state output voltage as a function of the $Na^+$ concentration, showing a cross-sensitivity of 44 mV $dec^{-1}$. The comparison between the OECT amplifier response in terms of $\Delta V_O$ vs. $c/c_{min}$ is displayed in Supplementary Fig. 5, demonstrating that a twofold variation in the $K^+$ concentration yields $\Delta V_O = 190$ mV, while the same $Na^+$ variation results in $\Delta V_O = 15$ mV. Importantly, the complementary OECT amplifier is able to sense a deviation of the $K^+$ concentration lower than the 20%, yielding an output variation of 45 mV when the concentration increases from 4.8 $10^{-3}$ M to 5.7 $10^{-3}$ M.

**Benchmarking transistor-based ion sensors.** To fairly compare the sensing performance of the OECT complementary amplifier with various transistor-based ion sensor technology platforms, we calculated the sensitivity normalized to the supply voltage $S_N = S_A V_{DD}^{-1}$. Figure 8a shows the sensitivity enhancement obtained by the OECT complementary amplifier with respect to the theoretical limit, which is given by the maximum sensitivity of a sensor normalized to the minimum supply voltage required to access the target range of concentration. Therefore, when the concentration range is equal to five orders of magnitude the theoretical limit results 200 $10^{-3}$ V $V^{-1}$ $dec^{-1}$. Interestingly, the maximum $S_N$ obtained by the proposed architecture is 2344 mV $V^{-1}$ $dec^{-1}$, more than one order of magnitude larger than the theoretical limit of transistor-based ion sensors.

In Fig. 8b the sensitivity and operating range of the ion selective OECT complementary amplifier is benchmarked against various transistor-based ion sensors devices and architectures, including silicon[10,12], porous-Si[14], graphene[11], zinc-oxide[13], amorphous In-Ga-ZnO[15], and organic materials technologies[16–19,43]. It shows that state-of-the-art ion sensors provide either wide operating range with small $S_N$, or average $S_N$ over a narrow range. Owing to the

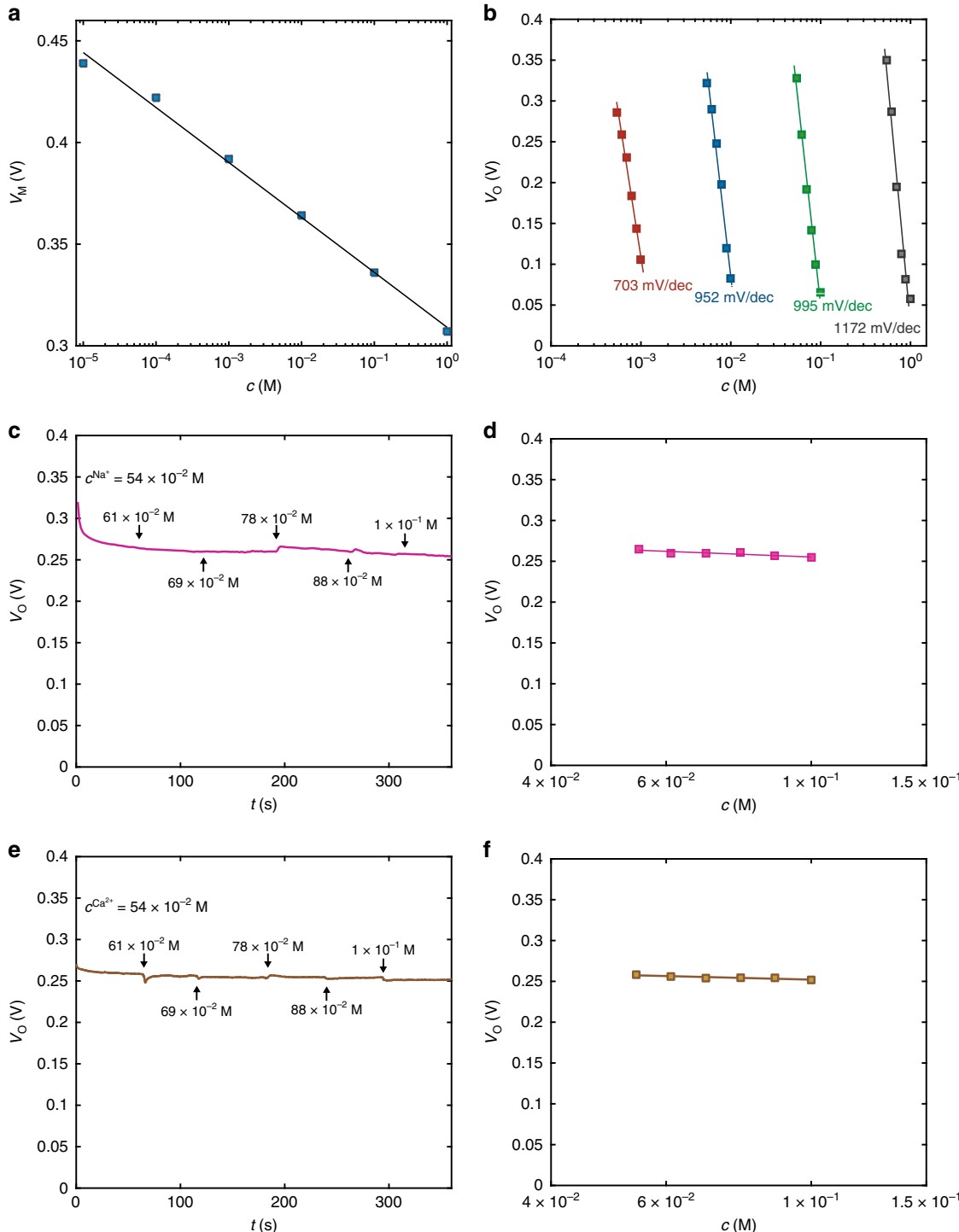

**Fig. 6 Selective ion detection and real time monitoring. a** Measured (symbols) $V_M$ as a function of the potassium concentration for the potassium selective organic electrochemical transistor complementary amplifier. The full line is the linear least square fit to the measurements, and its slope gives a wide-range sensitivity $S_M = 28$ mV dec$^{-1}$. **b** Measured (symbols) $V_O$ – $c^{K+}$ high sensitivity characteristics comprising the whole physiological range. Solid lines are the linear least square fit to the measurements, yielding $S_A = 703$ mV dec$^{-1}$, 952 mV dec$^{-1}$, 995 mV dec$^{-1}$, 1172 mV dec$^{-1}$. **c, e** Control experiment demonstrating the selectivity of the potassium-selective OECT complementary amplifier. $V_O$ is measured over time and the Na$^+$ (**c**) or Ca$^{2+}$ (**e**) concentration is systematically increased every 60 s from 5.4 10$^{-2}$ M to 1 10$^{-1}$ M. In both cases the K$^+$ concentration is constant and amounts to 5.4 10$^{-2}$ M. **d, f** Measured (symbols) steady state $V_O$ as a function of $c^{Na+}$ (**d**) or $c^{Ca2+}$ (**f**). Solid line is the linear least square fit to the measurements and yields a cross-sensitivity $S_{CR,Na+} = 33$ mV dec$^{-1}$ for Na$^+$, and $S_{CR,Ca2+} = 20$ mV dec$^{-1}$ for Ca$^{2+}$, confirming the excellent selectivity of the output response.

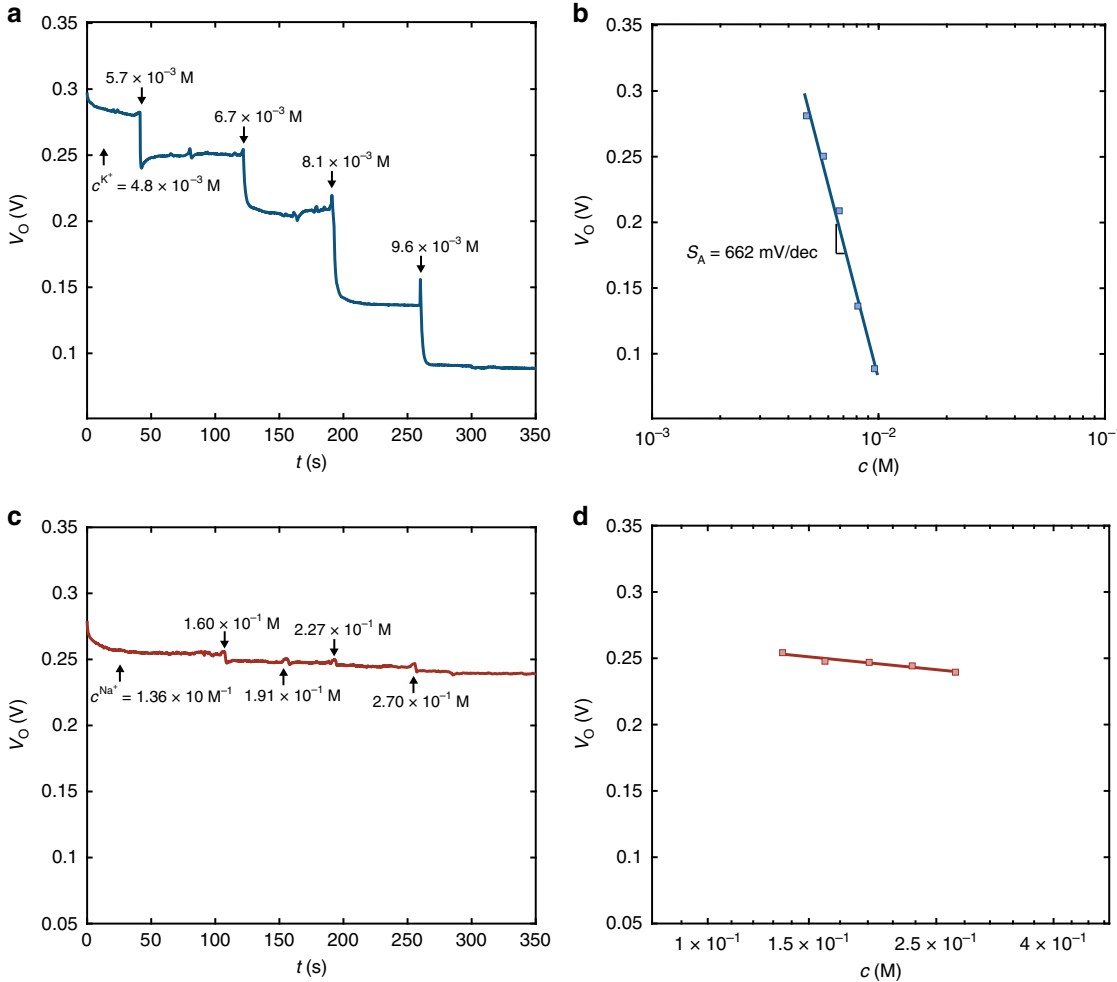

**Fig. 7 Selective K$^+$ sensing in human blood serum. a** Real time high sensitivity K$^+$ detection in human blood serum. The K$^+$ concentration range is 4.8 10$^{-3}$ M – 9.6 10$^{-3}$ M. The potassium concentration of the blood serum is increased over time by spiking small amounts of 1 10$^{-1}$ M KCl water solution. The measurements are performed at $V_I = 0.37$ V. **b** Measured steady-state output voltage $V_O$ (symbols) as a function of K$^+$ concentration $c$. Full blue line is the linear least square fit to the measurements and yields a sensitivity $S_A = 662$ mV dec$^{-1}$. **c** Control experiment demonstrating the selectivity against sodium in blood serum. $V_O$ is measured over time and the Na$^+$ concentration of the blood serum is systematically increased from 1.36 10$^{-1}$ M to 2.70 10$^{-1}$ M. The K$^+$ concentration is constant and amounts to 4.8 10$^{-3}$ M. **f** Measured (symbols) steady state $V_O$ as a function of $c^{Na+}$. Solid line is the linear least square fit to the measurements and yields a cross-sensitivity equal to 44 mV dec$^{-1}$, which is more than 15-fold lower than the sensitivity of the selected K$^+$ ions.

circuit-oriented multiscale approach, the OECT complementary amplifier device yields both superior sensitivity and wide range, providing easily-tunable ion-to-electron local transconduction and amplification. The performances are compared with more detail in Table 1. In addition to the multiscale high-sensitivity operation, the proposed OECT amplifier provides real time sensing capabilities and the supply voltage is amongst the lowest reported. As a result, the complementary OECT amplifier is able to selectively detect in real-time a variation of the K$^+$ concentration lower than 20% from the normal resting conditions, meeting the requirements for hypo- and hyperkalemia detection in blood serum[8].

## Discussion
Prospectively, the OECT complementary amplifier approach could be extended to a wide range of transistor-based bioelectronic technologies including for example, immunosensors[44] as well as metabolites[45–48], hormones[49], DNA[50], peptides[51], proteins[52] and viruses[53,54], detection. Along this direction, an emerging and rapidly growing research field is the single molecule detection with millimeter-sized transistors[55–59] where the biorecognition events result in a shift of the transistor threshold

voltage, which is typically of the order of few millivolts. The high-sensitivity, multiscale and reconfigurable operation provided by the OECT complementary amplifier can be extremely relevant also in this emerging research field to improve the signal-to-noise ratio, resolution and robustness, eventually achieving ultimately sensitive enhanced bioelectronics[60,61].

In conclusion, the proposed multiscale approach is a simple and effective way to capture the whole problem complexity without overlooking the smallest but meaningful details. Just like when we look at a big painting, first we have a global look at the context, then we get closer to appreciate a tiny detail, a hidden message or the brushstroke (Supplementary Fig. 6). In the same way, the ion sensitive OECT complementary amplifier provides ion sensing over wide range, i.e., the panoramic view of the problem, and high sensitivity detection, allowing to capture the tiny ion concentration variations that could be the meaningful fingerprint of pathological onsets. The top performing range and the unprecedented sensitivity, combined with real-time operation, tunable performances and low operating supply voltage, are enabled by a circuit-oriented device-aware sensor design approach. The proposed approach is general, and it can be extended to other sensing applications and transistor

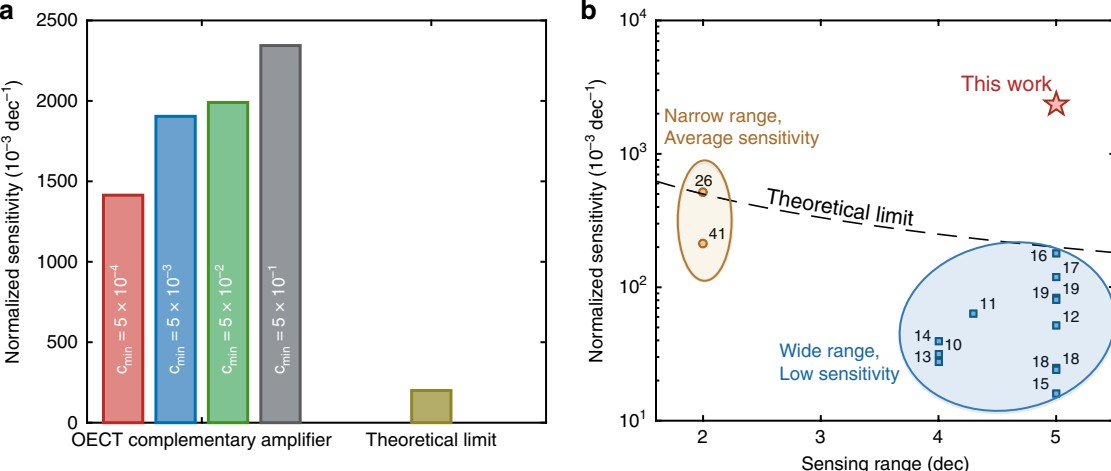

**Fig. 8 Sensing performance benchmark. a** Normalized Sensitivity $S_N$ of the organic electrochemical transistor complementary amplifier compared with the theoretical limit for ion detection over 5 orders of magnitude. The $S_N$ for the complementary OECT amplifier is equal to [1414,1904,1990,2344] mV V$^{-1}$ dec$^{-1}$ when $c_{min}$ is equal to [$5.4 \times 10^{-4}$ $5.4 \times 10^{-3}$ $5.4 \times 10^{-2}$ $5.4 \times 10^{-1}$] M, respectively, while the theoretical fundamental limit is equal to 200 mV V$^{-1}$ dec$^{-1}$. **b** Comparison between the normalized sensitivity and sensing range of the proposed OECT complementary amplifier with state-of-the-art ion sensors. It is worth to note that the $S_N$ is obtained as mV$^{-1}$ V$^{-1}$ dec$^{-1}$ when the output of the ion sensor is expressed as a voltage, while it is obtained as $\mu$A$^{-1}$ mA$^{-1}$ V when the output is a current. In both cases it yields 10$^{-3}$ dec$^{-1}$.

**Table 1 Sensing performances comparison.**

| Technology | Material | Selected ion | Supply voltage [V] | Ion concentration range [dec] | Normalized Sensitivity | | Real-time ion detection | Ref. |
|---|---|---|---|---|---|---|---|---|
| | | | | | Current [$\mu$A mA$^{-1}$ dec$^{-1}$] | Voltage [mV V$^{-1}$ dec$^{-1}$] | | |
| ISFET | Silicon | Na+, K+ | 1.8 | 4 | – | 31.6 | ✓ | 10 |
| ISFET | Graphene | K+ | 1.0 | 4.3 | – | 64.0 | ✓ | 11 |
| Hybrid ISFET | Silicon | K+ | 2.0 | 5 | – | 52.0 | ✓ | 12 |
| Single-gate TFT | ZnO | pH | 2.0 | 4 | – | 27.5 | – | 13 |
| double-gate TFT | a-IGZO | pH | 10.0 | 5 | – | 16.0 | – | 15 |
| Extended-gate FET | Porous-Si | Na+, K+ | 1.5 | 4 | 11 | 39.2 | – | 14 |
| EGOFET | P3HT | Na+ | 0.5 | 5 | 179 | 124.0 | ✓ | 16 |
| OECT | PEDOT:PSS | K+ | 0.4 | 5 | 72 | 120.0 | ✓ | 17 |
| Current-driven OECT | PEDOT:PSS | K+ | 0.8 | 2 | – | 517.5 | – | 26 |
| Current-driven OECT | IL-doped PEDOT:PSS | Na+ | 0.7 | 2 | – | 214.3 | – | 42 |
| Integrated IS-OECT | PEDOT:PSS | Na+ | 0.6 | 5 | – | 83.3 | – | 19 |
| Integrated IS-OECT | PEDOT:PSS | K+ | 0.6 | 5 | – | 80.0 | – | 19 |
| Membraneless OECT | p(T15c5-ran-EDOT) / PEDOT:PSS | Na+ | 1.2 | 5 | 25 | – | ✓ | 18 |
| Membraneless OECT | p(T18c6-ran-EDOT) / PEDOT:PSS | K+ | 1.2 | 4 | 24 | – | ✓ | 18 |
| OECT complementary amplifier | PEDOT: PSS, BBL | K+ | 0.5 | 5 | – | 2344.0 | ✓ | This work |

The table compares the performances of several transistor-based sensor technologies by considering the type of ion, the supply voltage ($V_{DD}$), the sensing range, the normalized sensitivity $S_N$ and the real-time detection capability.

technologies, paving the way to enhanced multifunctional ion detection and opening opportunities for high-performance bioelectronics.

## Methods

**Device fabrication**. OECTs were fabricated following the procedure described in Ref. [62]. Briefly, gold source and drain electrodes were deposited by sputtering, using chromium as adhesion layer, on a $26 \times 76$ mm$^2$ glass slide. The electrodes were photolithographically patterned, defining the channels dimensions. The p-type

channel width ($W$) and length ($L$) are 200 $\mu$m and 50 $\mu$m, respectively. The n-type devices interdigitated electrodes defined $W = 80$ mm, $L = 20$ $\mu$m. A first layer of parylene-C is deposited to isolate the contact lines from the electrolyte, and a second sacrificial layer is deposited on top of it to enable polymer patterning with a peel-off procedure. Soap is used to separate the two parylene-C layers, enabling the peel-off of the sacrificial layer. PEDOT:PSS (Clevios PH 1000) was mixed with 5% ethylene glycol, 0.1 wt% dodecyl benzene sulf acid and 1 wt% of (3-glycidyloxypropyl) trimethoxysilane, and spin coated at 1000 rpm for 60 s. The devices were baked at 140 °C for 1 h. BBL was mixed with Methanesulfonic Acid (5 mg/ml) and stirred overnight at 70 °C. The obtained solution was spin coated at 400 rpm for

60 s, then the glass slide was soaked in DI water for 1 h. The devices were baked at 140 °C for 4 h.

**Ion selective membranes fabrication**. High molecular weight PVC (36.5 wt.%) was mixed with potassium ionophore III (2.5 wt.%), potassium tetrakis(4-chlorophenyl-)borate (0.5 wt.%) and diisodecyl adipate (60.5 wt.%) in tetrahydrofuran THF (500 mg/5 mL). The mixture was spin coated at 300 rpm for 120 s. A rubber ring defined the membrane dimension and provided mechanical stability. ISM-OECT: a PDMS well was used to confine the filling solution (1 M NaCl) and provided mechanical support to the membrane. A small well is placed on top of the stack to confine the analyte.

**Electrical characterization**. The measurements were performed using KCl or NaCl at several concentrations. Human blood serum was purchased from Sigma-Aldrich and used as received, with a starting KCl concentration of $5 \ 10^{-3}$ M. A tin wire was used for PEDOT:PSS OECT gate electrode, while Ag/AgCl was used for the BBL OECT gate. The electrical characteristics were measured with two Keithley 2636 A. Ad-hoc microfluidics are designed to provide electrical connection to the device and a housing for the analyte (See Supplementary Fig. 7). The OECT complementary amplifier transfer characteristics were measured by sweeping the input voltage from 0 V to $V_{DD}$ with a rate of 10 mV/s.

## Data availability
The data that support the findings of this study are available from the corresponding author on reasonable request.

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

## Acknowledgements

We are thankful to Prof. Simone Fabiano and Dr. Hengda Sun for their valuable advices on the BBL semiconductor processing. The authors Z.M.K.-V. and F.T. acknowledge the financial support of the European Commission for the project SiMBiT (Horizon 2020 ICT, contract number 824946) and the financial support of the European Union, Italian Government and Lombardia Region for the project BIOSCREEN (POR FESR 2014-2020, ID number 1831459, CUP E81B20000320007).

## Author contributions

P.R. fabricated and measured the OECTs, ion selective membranes, and amplifiers. P.G. supported and supervised the fabrications and measurements. D.K. fabricated the OECTs. K.L. supported the device fabrication. P.W.M.B., Z.M.K.V. and F.T. supervised the project. F.T. conceived the idea and designed the experiments. P.R. and F.T. wrote the paper. All the authors revised and commented on the paper.

## Competing interests

The authors declare no competing interests.
