## [Peer Review File · Nature Communications]

Reviewers' Comments:

Reviewer #1:

Remarks to the Author:

Comments

The paper written by Paolo Rmele et al. describes complementary organic electrochemical transistors amplifier by connecting p-type and n-type OECT for real time detection of ion concentration. The authors showed somehow acceptable sensing performance depending on the concentration of ion, however it is difficult for the reviewer to find novelty and ground-breaking compared to the other researches. Thus, the reviewer suggests that this manuscript is not suitable for the publication and recommends submission of this article to more specialized journals. Here are some comments for this paper.

- 1) This article introduces the complementary organic electrochemical transistors amplifier for multiscale real time ion detection. However, there are many researches about ion-sensitive organic electronics as mentioned in main context. The reviewer feels that the authors should consider practical and feasible demo- and application in biomedical field for novelty of this research.
- 2) In this article, selective ion detection based on the complementary OECTs amplifier in Figure 6 with potassium (K^+) and sodium ion (Na^+) which are components of physiological system. The selectivity of the amplifier is described so well by real time monitoring, however, there are other ions in physiological system such as calcium ion (Ca^{2+}) having higher charge compared to ions mentioned above. So, the reviewer feels that the authors should add experimental results from other ions having the charge more than +1 for better demonstration.
- 3) The authors show higher sensitivity of the complementary OECTs amplifier in Figure 4 than that of a single OECT with PEDOT:PSS. However, there is no obvious explanation about the relationship between concentration of ions and sensitivity that the higher concentration results in the larger value of sensitivity in Figure 4c. The authors should supply more detail about it.
- 4) The authors successfully construct the complementary OECTs amplifier with each p-type and n-type OECT. Especially, p-type OECT is operated by depletion mode from compensation of the effect of anionic sulfonate groups on the PSS. However, complementary logic circuits with both p- and n-type OECT by accumulation mode (Adv. Mater. 2018, 30, 1704916). It would be better if there are specific reasons that why the authors utilized the depletion mode OECT in this paper.

Reviewer #2:

Remarks to the Author:

The paper regards the realization of an OECT transistor amplifiers, based on two OECT transistor one p type based on PEDOT:PSS and one n type based on BBL. The sensing of ions in solution is a well-known application for these devices, but the realization of an amplifier, is an interesting development, because it allows to explore the potential advantage of these devices. In particular, the authors show a high sensitivity of the complementary amplifiers, that overcome the fundamental limit. These results are original and shows all the potential of OECT, as part of a logic system, with their characteristics of large transconductance. The paper is well written and clear. The case of an ion selective membrane is also showed to enhance the selectivity of the device. The data are convincing and I suggest to publish the paper, because it could be a reference for the sensitivity of these devices. Maybe an actual image of the device in the supporting is useful.

Reviewer #3:

None

Response Letter

We thank the Reviewers for their valuable comments and suggestions. Herewith we provide the point-by-point responses to the Reviewers. All the new results, analysis and changes are highlighted in the revised manuscript.

Reviewer #1 (Remarks to the Author):

The paper written by Paolo Romele et al. describes complementary organic electrochemical transistors amplifier by connecting p-type and n-type OECT for real time detection of ion concentration. The authors showed somehow acceptable sensing performance depending on the concentration of ion, however it is difficult for the reviewer to find novelty and ground-breaking compared to the other researches. Thus, the reviewer suggests that this manuscript is not suitable for the publication and recommends submission of this article to more specialized journals. Here are some comments for this paper.

Authors: We are sorry that the Reviewer did not find our work suitable for publication. We agree with the Reviewer that the in-situ quantification of the ion concentration in aqueous media has been extensively addressed by ion sensors based on both field-effect and electrochemical transistors fabricated with a wide palette of materials including for example silicon, graphene, nanowires, metal-oxides and organics. Nevertheless, the widespread adoption of ion sensors is currently hampered by the intrinsic trade-off between sensitivity, ion concentration range and operating voltage. Even more importantly, the possibility to detect both the ion concentration in a wide range and track small variations of the ion concentration with respect to the detected concentration – viz. multiscale high-sensitivity ion detection – is still an open challenge.

In our work we show simultaneous multiscale real-time and high-sensitivity ion detection with complementary organic electrochemical transistors amplifiers. The ion-sensing amplifier integrates in the same device both selective ion-to-electron transduction and local signal amplification demonstrating selective real-time ion detection with a sensitivity up to 1172 mV dec^{-1} at a supply voltage equal to 0.5 V and with an operative range of concentrations equal to $10^{-5} \text{ M} - 1 \text{ M}$. This results in a voltage sensitivity normalized to the supply voltage larger than $2300 \text{ mV V}^{-1} \text{ dec}^{-1}$, which is the highest value ever reported overcoming the fundamental limit of conventional methods. Therefore, the ion-sensing amplifiers provide both ion detection over a range of five orders of magnitude and real-time monitoring of variations two orders of magnitude lower than the detected concentration, viz. multiscale ion detection. We note that the proposed approach is general and can be extended to any transistor technology.

The performance of the proposed approach is systematically compared with the state of the art in Fig. 8 and Table 1. To fairly compare the sensing performance of the OECT complementary amplifier with various transistor-based ion sensor technology platforms, we calculated the sensitivity normalized to the supply voltage (S_N). Fig. 8a shows the sensitivity enhancement obtained with the OECT complementary amplifier with respect to the theoretical limit, which is given by the maximum sensitivity of a sensor normalized to the minimum supply voltage required to access the target range of concentration. Therefore, when the concentration range is equal to five orders of magnitude the resulting theoretical limit amounts to $200 \text{ mV V}^{-1} \text{ dec}^{-1}$. Interestingly, the maximum normalized sensitivity obtained with the proposed architecture is $2344 \text{ mV V}^{-1} \text{ dec}^{-1}$, more than one order of magnitude larger than the theoretical limit of transistor-based ion sensors.

In Fig. 8b the sensitivity and operating range of the ion selective OECT complementary amplifier is benchmarked against various transistor-based ion sensors devices and architectures, including silicon^[10,12], porous-Si^[14], graphene^[11], zinc-oxide^[13], amorphous In-Ga-ZnO^[15], and organic materials technologies^[16–19,40].

It shows that state-of-the-art ion sensors provide either wide operating range with small S_N , or average S_N over a narrow range. Owing to the circuit-oriented multiscale approach, the proposed OECT complementary amplifier device yields both superior sensitivity and wide range, providing easily-tunable ion-to-electron local transconduction and amplification. The performances of the various sensors are compared in more detail in Table 1. In addition to the multiscale high-sensitivity operation, the proposed OECT amplifier provides real time record sensing capabilities and the supply voltage is amongst the lowest reported. As a result, the complementary OECT amplifier is able to selectively sense a deviation of the K^+ concentration lower than the 20% from the normal condition in real-time, meeting the requirements for hypo- and hyperkalemia detection in blood serum^[8].

In addition, according to the Reviewer suggestions, we included in the revised manuscript a demonstration of selective real-time ion monitoring in human blood serum, which is an extremely relevant biological fluid for ion sensing applications and encompasses the complexity typical of biological environments. We demonstrate that the proposed architecture can be used in such environment retaining its unparalleled ion-sensing performances with good selectivity, enabling high sensitivity real-time ion detection of physiologically relevant ions and allowing to meet the challenging performances required for the rapid detection of pathological states. As a meaningful example, we consider the potassium concentration in human blood serum, that is normally in the range of $3.5 \cdot 10^{-3}$ to $5.5 \cdot 10^{-3}$ M. A departure from this range lower than 20% ($\approx 1 \cdot 10^{-3}$ M) can be associated to severe pathological states^[8]. To the best of our knowledge, no state-of-the-art transistor-based ion sensor is able to detect such a small variation. The new results included in the revised manuscript show that the proposed ion sensitive OECT amplifier meets this requirement, providing the unprecedented sensitivity of up to 662 mV dec^{-1} to ion concentration variations in real-time. Importantly, this performance is enabled by a device-aware design approach, exploiting the characteristic features of OECTs to achieve high performances.

The quality of the manuscript has been further enhanced by the addition of all the analyses suggested by the Reviewer, providing meaningful insight on the design, working principles and performances of the complementary OECT amplifier. Thus, we believe that the revised work could be interesting for several fields, inspiring the community working on OECT sensors and circuits with a new design paradigm, and addressing the need for high performance biosensing devices of the biomedical community.

According to the Reviewer comments we added these results, explanations and analysis in the revised manuscript.

Fig. 8: Sensing performance benchmark. **a** Normalized Sensitivity S_N of the OECT complementary amplifier compared to the theoretical limit for ion detection over 5 orders of magnitude. The S_N for the complementary OECT amplifier is equal to [1414, 1904, 1990, 2344] $\text{mV V}^{-1} \text{ dec}^{-1}$ when c_{\min} is equal to [5.4×10^{-4} 5.4×10^{-3} 5.4×10^{-2} 5.4×10^{-1}] M,

respectively, while the theoretical fundamental limit is equal to $200 \text{ mV V}^{-1} \text{ dec}^{-1}$. **b** Comparison between the normalized sensitivity and sensing range of the proposed OECT complementary amplifier with state-of-the-art ion sensors. It is worth to note that the S_N is obtained as $\text{mV}^{-1} \text{ V}^{-1} \text{ dec}^{-1}$ when the output of the ion sensor is expressed as a voltage, while it is obtained as $\mu\text{A}^{-1} \text{ mA}^{-1} \text{ V}$ when the output is a current. In both cases it yields 10^{-3} dec^{-1} .

Technology	Material	Selected ion	Supply voltage [V]	Ion concentration range [dec]	Normalized Sensitivity		Real-time ion detection	Ref.
					Current [$\mu\text{A mA}^{-1} \text{ dec}^{-1}$]	Voltage [$\text{mV V}^{-1} \text{ dec}^{-1}$]		
ISFET	Silicon	Na+, K+	1.8	4	-	31.6	✓	[10]
ISFET	Graphene	K+	1.0	4.3	-	64.0	✓	[11]
Hybrid ISFET	Silicon	K+	2.0	5	-	52.0	✓	[12]
Single-gate TFT	ZnO	pH	2.0	4	-	27.5	-	[13]
double-gate TFT	a-IGZO	pH	10.0	5	-	16.0	-	[15]
Extended-gate FET	Porous-Si	Na+, K+	1.5	4	11	39.2	-	[14]
EGOFET	P3HT	Na+	0.5	5	179	124.0	✓	[16]
OECT	PEDOT:PSS	K+	0.4	5	72	120.0	✓	[17]
Current-driven OECT	PEDOT:PSS	K+	0.8	2	-	517.5	-	[26]
Current-driven OECT	IL-doped PEDOT:PSS	Na+	0.7	2	-	214.3	-	[40]
Integrated IS-OECT	PEDOT:PSS	Na+	0.6	5	-	83.3	-	[19]
Integrated IS-OECT	PEDOT:PSS	K+	0.6	5	-	80.0	-	[19]
Membraneless OECT	p(T15c5-ran-EDOT) / PEDOT:PSS	Na+	1.2	5	25	-	✓	[18]
Membraneless OECT	p(T18c6-ran-EDOT) / PEDOT:PSS	K+	1.2	4	24	-	✓	[18]
OECT complementary amplifier	PEDOT:PSS, BBL	K+	0.5	5	-	2344.0	✓	This work

Table 1: Sensing performances comparison. The table compares the performances of several transistor-based sensor technologies by considering the type of ion, the supply voltage (V_{DD}), the sensing range, the normalized sensitivity S_N and the real-time detection capability.

Fig. 7: selective K^+ sensing in human blood serum. **a** Real time high sensitivity K^+ detection in human blood serum. The K^+ concentration range is $4.8 \cdot 10^{-3} \text{ M} - 9.6 \cdot 10^{-3} \text{ M}$. The potassium concentration of the blood serum is increased over time by spiking with a $1 \cdot 10^{-1} \text{ M}$ KCl water solution. The measurements are performed at $V_i = 0.37 \text{ V}$. **b** Measured steady-state output voltage V_o (symbols) as a function of K^+ concentration c . Full blue line is the linear least square fit to the measurements and yields a sensitivity $S_A = 662 \text{ mV dec}^{-1}$. **c** Control experiment demonstrating the selectivity against sodium in blood serum. V_o is measured over time and the blood serum Na^+ concentration is systematically increased from $1.36 \cdot 10^{-1} \text{ M}$ to $2.70 \cdot 10^{-1} \text{ M}$. The K^+ concentration is constant and amounts to $4.8 \cdot 10^{-3} \text{ M}$. **f** Measured (symbols) steady state V_o as a function of c^{Na^+} . Solid line is the linear least square fit to the measurements and yields a cross-sensitivity equal to 44 mV dec^{-1} , which is more than 15-fold lower than the sensitivity to the selected K^+ ions.

References

- [8] Rastegar, A. Serum potassium. In: Walker, H.K., Hall, W.D. & Hurst, J.W. (Eds.). Clinical methods: The history, physical, and laboratory examinations. 3rd ed. Boston, MA: Butterworths; 1990.
- [10] Zhang, J. et al. Sweat Biomarker Sensor Incorporating Picowatt, Three-Dimensional Extended Metal Gate Ion Sensitive Field Effect Transistors. ACS Sens. 4, 2039–2047 (2019).
- [11] Li, H. et. al. Graphene field effect transistors for highly sensitive and selective detection of K^+ ions. Sens. Actuators B Chem. 253, 759–765 (2017).
- [12] Bao, C., Kaur, M. & Kim, W. S. Toward a highly selective artificial saliva sensor using printed hybrid field effect transistors. Sens Actuators B Chem 285, 186–192 (2019).

- [13] Yano, M. et al. Zinc oxide ion-sensitive field-effect transistors and biosensors. *Phys. Status Solidi Appl. Mater. Sci.* 211, 2098–2104 (2014).
- [14] Kabaa, E. A., Abdulateef, S. A., Ahmed, N. M., Hassan, Z. & Sabah, F. A. A novel porous silicon multi-ions selective electrode based extended gate field effect transistor for sodium, potassium, calcium, and magnesium sensor. *Appl. Phys. A* 125, 753 (2019).
- [15] Kumar, N., Kumar, J. & Panda, S. Back-channel electrolyte-gated a-IGZO dual-gate thin-film transistor for enhancement of pH sensitivity over nernst limit. *IEEE Electron Device Lett.* 37, 500–503 (2016).
- [16] Schmoltner, K., Kofler, J., Klug, A. & List-Kratochvil, E. J. W. Electrolyte-gated organic field-effect transistor for selective reversible ion detection. *Adv. Mater.* 25, 6895–6899 (2013).
- [17] Sessolo, M., Rivnay, J., Bandiello, E., Malliaras, G. G. & Bolink, H. J. Ion- selective organic electrochemical transistors. *Adv. Mater.* 26, 4803–4807 (2014).
- [18] Wustoni, S. et al. Membrane-Free Detection of Metal Cations with an Organic Electrochemical Transistor. *Adv. Funct. Mater.* 1904403 (2019).
- [19] Pierre, A., Doris, S. E., Lujan, R., Street, R. A., Monolithic Integration of Ion-Selective Organic Electrochemical Transistors with Thin Film Transistors on Flexible Substrates. *Adv. Mater. Technol.* 4, 1800577 (2019).
- [26] Ghittorelli, M. et al. High-sensitivity ion detection at low voltages with current-driven organic electrochemical transistors. *Nat. Commun.* 9, 1441 (2018).
- [40] Wu, X. et al. Ionic-Liquid Doping Enables High Transconductance, Fast Response Time, and High Ion Sensitivity in Organic Electrochemical Transistors. *Adv. Mater* 31,1805544 (2019).

1) This article introduces the complementary organic electrochemical transistors amplifier for multiscale real time ion detection. However, there are many researches about ion-sensitive organic electronics as mentioned in main context. The reviewer feels that the authors should consider practical and feasible demo- and application in biomedical field for novelty of this research.

[R1Q1] We thank the Reviewer for his/her valuable suggestion. We agree that a practical and feasible application of the amplifier further demonstrates the relevance of our work for the biomedical field. According to the Reviewer’s comment, we performed real time detection of potassium in human blood serum, representing a meaningful biological fluid for ion detection. The normal range for serum potassium concentration is $3.5 \cdot 10^{-3}$ to $5.5 \cdot 10^{-3}$ M, and a minor departure from this range, even lower than 20%, is associated to hypo- or hyperkalemia, leading to severe morbidity and mortality⁸. Importantly, such a small variation has to be sensed in a complex environment including, amongst others, a sodium concentration in the range of $1.35 \cdot 10^{-1}$ to $1.45 \cdot 10^{-1} \text{ M}^{[A1]}$, which is more than 1 order of magnitude larger than the potassium physiological concentration. Therefore, the real-time monitoring of potassium blood serum is a major challenge, and it is of paramount importance for the rapid detection and treatment of pathological states.

The real-time monitoring of potassium in blood serum with a complementary OECT amplifier is displayed in Fig. 7a. The output voltage is continuously measured over time by varying the potassium concentration from $4.8 \cdot 10^{-3}$ M to $9.6 \cdot 10^{-3}$ M, which are K^+ concentrations representative of healthy and pathological states. The K^+ concentration of the blood serum is varied by spiking the analyte with a small amount of $1 \cdot 10^{-1}$ M KCl solution. The background sodium concentration is $1.36 \cdot 10^{-1}$ M, i.e. about 2 orders of magnitude larger than the K^+ concentration. Fig. 7a shows that every time the analyte K^+ concentration is increased, the output voltage readily decreases. The measured steady state V_0 as a function of c is shown in Fig. 7b. The least square linear approximation of the V_0 - c characteristic yields an average sensitivity equal to 662 mV dec^{-1} , which is in agreement with that obtained with a KCl water solution in the same range of c (Fig. 6b). The selectivity control experiment is displayed in Fig. 7c, where the starting Na^+ concentration of $1.36 \cdot 10^{-1}$ M is

increased over time up to $2.72 \cdot 10^{-1}$ M by spiking with a 5 M NaCl solution. In this control experiment a negligible variation of the measured output voltage is obtained. Fig. 7d displays the measured steady state output voltage as a function of the Na^+ concentration, showing a cross-sensitivity of 44 mV dec^{-1} . The comparison between the OECT amplifier response in terms of ΔV_O vs. c/c_{\min} is displayed in Supplementary Figure 4, showing that a two-fold variation in the K^+ concentration yields $\Delta V_O = 190 \text{ mV}$, while the same Na^+ variation results in $\Delta V_O = 15 \text{ mV}$. Importantly, the complementary OECT amplifier is able to sense deviation of the K^+ concentration lower than 20%, yielding an output variation of 45 mV when the selected K^+ concentration increases from $4.8 \cdot 10^{-3}$ M to $5.7 \cdot 10^{-3}$ M. The sensitivity to K^+ variations is more than 15 times the sensitivity to Na^+ variations, meeting the sensitivity and selectivity requirements for hypo- and hyperkalemia detection in blood serum^[8].

According to the Reviewer's suggestion, we included these results and analysis in the revised manuscript.

Fig. 7: selective K^+ sensing in human blood serum. **a** Real time high sensitivity K^+ detection in human blood serum. The K^+ concentration range is $4.8 \cdot 10^{-3}$ M – $9.6 \cdot 10^{-3}$ M. The potassium concentration of the blood serum is increased over time by spiking with a $1 \cdot 10^{-1}$ M KCl water solution. The measurements are performed at $V_I = 0.37$ V. **b** Measured steady-state output voltage V_O (symbols) as a function of K^+ concentration c . Full blue line is the linear least square fit to the measurements and yields a sensitivity $S_A = 662 \text{ mV dec}^{-1}$. **c** Control experiment demonstrating the selectivity against sodium in blood serum. V_O is measured over time and the blood serum Na^+ concentration is systematically increased from $1.36 \cdot 10^{-1}$ M to $2.70 \cdot 10^{-1}$ M. The K^+ concentration is constant and amounts to $4.8 \cdot 10^{-3}$ M. **f** Measured (symbols) steady state V_O as a function of c^{Na^+} . Solid line is the linear least square fit to the measurements and yields a cross-sensitivity equal to 44 mV dec^{-1} , which is more than 15-fold lower than the sensitivity to the selected K^+ ions.

Supplementary Figure 4: Blood serum selective ion detection. Measured (symbols) output voltage variation $\Delta V_0 = V_0(c) - V_0(c_{\min})$ as a function of c/c_{\min} in blood serum sample. The OECT complementary amplifier is selective to K^+ ions. Blue symbols are measured with $c_{\min}^{K^+} = 4.8 \cdot 10^{-3}$ M. Red symbols are the control experiment measured with $c_{\min}^{Na^+} = 1.36 \cdot 10^{-1}$ M. Dotted lines are guides for the eye.

References

- [A1] B. G. Fahy, J. T. Murphy & J. L. Atlee. Disorders of Water Homeostasis: Hyponatremia and Hypernatremia. Book Chapter in Complications in Anesthesia (Second Edition), Edited by J. L. Atlee, W. B. Saunders. 429-435 (2007)
- [8] Rastegar, A. Serum potassium. In: Walker, H.K., Hall, W.D. & Hurst, J.W. (Eds.). Clinical methods: The history, physical, and laboratory examinations. 3rd ed. Boston, MA: Butterworths; 1990.

2) In this article, selective ion detection based on the complementary OECTs amplifier in Figure 6 with potassium (K^+) and sodium ion (Na^+) which are components of physiological system. The selectivity of the amplifier is described so well by real time monitoring, however, there are other ions in physiological system such as calcium ion (Ca^{2+}) having higher charge compared to ions mentioned above. So, the reviewer feels that the authors should add experimental results from other ions having the charge more than +1 for better demonstration.

[R1Q2] According to the Reviewer's suggestion we extended the real-time analysis including calcium ion (Ca^{2+}). The selectivity of the OECT amplifier against Ca^{2+} is displayed in Fig. 6e of the revised manuscript. In this control experiment, a K^+ -selective membrane is integrated in the complementary OECT amplifier, the K^+ concentration is constant and amounts to $5.4 \cdot 10^{-2}$ M, while the Ca^{2+} concentration is systematically increased from $5.4 \cdot 10^{-2}$ M to $1 \cdot 10^{-1}$ M. The measured output voltage V_0 as function of time is independent of the Ca^{2+} spiked to the electrolyte solution (Fig. 6e). As a further confirmation, in Fig. 6f the measured steady state V_0 is displayed as a function of the Ca^{2+} concentration. A negligible response is triggered by the Ca^{2+} concentration and the cross-sensitivity amounts to 20 mV dec^{-1} , which is about 50-folds lower than the K^+ sensitivity in the same range of concentrations (i.e. 995 mV dec^{-1}) and 35-folds lower than the minimum K^+ sensitivity (i.e. 703 mV dec^{-1} in the range $5 \cdot 10^{-4} - 1 \cdot 10^{-3}$ M) (Supplementary Figure 3).

We thank the Reviewer for his/her comment, and we included the analysis and results in the revised manuscript.

3) The authors show higher sensitivity of the complementary OECTs amplifier in Figure 4 than that of a single OECT with PEDOT:PSS. However, there is no obvious explanation about the relationship between concentration of ions and sensitivity that the higher concentration results in the larger value of sensitivity in Figure 4c. The authors should supply more detail about it.

[R1Q3] We thank the Reviewer for pointing out that the relationship between the concentration of ions and the sensitivity should be better addressed.

The real time sensitivity of the amplifier reported in Fig. 4 is given by $S_A = dV_O/dc = A S_M$ (Eq. 3 of the manuscript). Supplementary Figure 2a shows that the amplification $A = dV_O/dV_I$ of the complementary OECT amplifier increases by increasing ion concentration, yielding a higher S_A , and this can be explained as follows. The amplification A at the transition voltage V_M reads (Eq. (4) of the manuscript):

$$A(V_M) = \frac{1}{I_D(V_M)} \frac{g_{mn} \left[1 + \lambda_n \frac{V_{DD}}{2} \right] + g_{mp} \left[1 + \lambda_p \frac{V_{DD}}{2} \right]}{\lambda_n + \lambda_p} \quad (4)$$

where $I_D(V_M)$ is the current flowing through the OECTs at $V_I = V_M$, V_{DD} is the supply voltage, g_{mn} and g_{mp} are the n-type and p-type OECTs transconductances, respectively, and λ_n , λ_p account for the channel length modulation in the n-type and p-type OECTs, respectively. According to Eq. (4) the amplification is inversely proportional to $I_D(V_M)$. Supplementary Figure 2b shows that $I_D(V_M)$ progressively decreases by increasing the ion concentration. The lower $I_D(V_M)$ can be explained by considering that the OECT threshold voltage V_{Tp} shift to lower voltages by increasing the ion concentration, and this yields a lower I_D when $V_I = V_M$. As a consequence, higher concentrations result in larger A and, in turn, in a larger sensitivity.

According to the Reviewer's suggestion we added this additional analysis in the revised manuscript.

Supplementary Figure 2: Relationship between amplification and ion concentration. **a** Measured amplification A of the complementary OECT amplifier at various ion concentrations. From the rightmost to the leftmost characteristic the ion concentration is equal to $[10^{-5}, 10^{-4}, 10^{-3}, 10^{-2}, 10^{-1}, 1]$ M. **b** Measured I_D of the OECT complementary amplifier at various ion concentrations. From the lightest to the darkest characteristic the ion concentration is equal to $[10^{-5}, 10^{-4}, 10^{-3}, 10^{-2}, 10^{-1}, 1]$ M.

4) The authors successfully construct the complementary OECTs amplifier with each p-type and n-type OECT. Especially, p-type OECT is operated by depletion mode from compensation of the effect of anionic sulfonate groups on the PSS. However, complementary logic circuits with both p- and n-type OECT by accumulation mode (*Adv. Mater.* 2018, 30, 1704916). It would be better if there are specific reasons that why the authors utilized the depletion mode OECT in this paper.

[R1Q4] We thank the Reviewer for his/her comment. The OECT materials have a strong impact on the ion sensitive performances of the proposed complementary OECT amplifier. More in detail, the product of the material volumetric capacitance and mobility (μC_V) determines the OECT transconductance³³ (i.e. g_{mn} , g_{mp}), which in turn affects the amplification parameter A and thus its real time sensitivity S_A (see Eq. (4)). PEDOT:PSS shows top performing μC_V product among the OECT materials³³, ensuring high S_A . Furthermore, the material ionic sensitivity in terms of $\partial V_{Tp}/\partial c$ affects the S_M parameter. The ionic response of PEDOT:PSS OECTs is well studied in the literature and this allows to precisely design the characteristics of the ion sensor. For these reasons we chose PEDOT:PSS as the p-type material for the OECT amplifier.

We note that the proposed architecture is general, and in the case OECT materials with improved performances and functionalities will be developed and studied in the future, they could eventually be integrated in the complementary OECT amplifier.

According to the Reviewer's suggestions, we included this explanation in the revised manuscript.

References

[33] Inal, S., Malliaras, G. G. & Rivnay, J. Benchmarking organic mixed conductors for transistors. *Nat. Commun.* 8, 1767 (2017).

Reviewer #2 (Remarks to the Author):

The paper regards the realization of an OECT transistor amplifiers, based on two OECT transistor one p type based on PEDOT:PSS and one n type based on BBL. The sensing of ions in solution is a well-known application for these devices, but the realization of an amplifier, is an interesting development, because it allows to explore the potential advantage of these devices. In particular, the authors show a high sensitivity of the complementary amplifiers, that overcome the fundamental limit. These results are original and shows all the potential of OECT, as part of a logic system, with their characteristics of large transconductance. The paper is well written and clear. The case of an ion selective membrane is also showed to enhance the selectivity of the device. The data are convincing and I suggest to publish the paper, because it could be a reference for the sensitivity of these devices. Maybe an actual image of the device in the supporting is useful.

Authors [R2Q1]: We thank the Reviewer for his/her comments. We are glad that he/she recommended the work for publication in Nature Communications. According to the Reviewer's suggestion we added a picture of the device to the Supplementary Information.

Supplementary Figure 6: Image of a complementary OECT amplifier. The fabricated complementary OECT amplifier is hosted into a custom 3D printed microfluidic that provides reliable electrical connection with the measurement instruments and a well containing the analyte.

Reviewers' Comments:

Reviewer #1:

Remarks to the Author:

Comments

Thank you for your responses to our comments by adding supporting data and explanation. The reviewer suggests that this paper would be acceptable for the publication after revision of followings.

- 1) This reviewer still concerned about the novelty of the work. Although this work appeals sensing performance, complimentary OECT amplifier does not look new. Please address recent history on the development of OECT amplifier devices.
- 2) Please check the number on the x-axis of Fig. 6d, the reviewer thinks that authors need to change 1×10^1 to 1×10^{-1} .

Reviewer #2:

Remarks to the Author:

The authors have modified the paper as suggested
I recommend the publication with this modifications

Response Letter

Reviewer #1 (Remarks to the Author): *Thank you for your responses to our comments by adding supporting data and explanation. The reviewer suggests that this paper would be acceptable for the publication after revision of followings.*

1) This reviewer still concerned about the novelty of the work. Although this work appeals sensing performance, complimentary OECT amplifier does not look new. Please address recent history on the development of OECT amplifier devices.

2) Please check the number on the x-axis of Fig. 6d, the reviewer thinks that authors need to change 1×10^1 to 1×10^{-1} .

Authors: We are glad that the Reviewer found the revised version of our work acceptable for the publication. We further revised the manuscript addressing recent history on the development of OECT amplifier devices [33,34]. Moreover, we fixed the error in the label of Fig. 6d. We thank the Reviewer for his/her comments and suggestions.

[33] Venkatraman, V., et al. Subthreshold Operation of Organic Electrochemical Transistors for Biosignal Amplification. Adv. Sci. 5, 1800453 (2018).

[34] Braendlein, M., Lonjaret, T., Leleux, P., Badier, J. M. & Malliaras, G.G.. Voltage Amplifier Based on Organic Electrochemical Transistor. Adv. Sci. 4, 1600247 (2017).

Reviewer #2 (Remarks to the Author): *The authors have modified the paper as suggested I recommend the publication with this modifications.*

Authors: We thank the Reviewer for his/her suggestions and recommendations.